# Deciphering the Role of Epstein–Barr Virus Latent Membrane Protein 1 in Immune Modulation: A Multifaced Signalling Perspective

**DOI:** 10.3390/v16040564

**Published:** 2024-04-04

**Authors:** Petra Šimičić, Margarita Batović, Anita Stojanović Marković, Snjezana Židovec-Lepej

**Affiliations:** 1Department of Oncology and Nuclear Medicine, Sestre Milosrdnice University Hospital Center, Vinogradska cesta 29, 10 000 Zagreb, Croatia; petrasimicic@gmail.com; 2Department of Clinical Microbiology and Hospital Infections, Dubrava University Hospital, Avenija Gojka Šuška 6, 10 000 Zagreb, Croatia; margi.batovic@gmail.com; 3Department of Immunological and Molecular Diagnostics, University Hospital for Infectious Diseases “Dr. Fran Mihaljević”, Mirogojska 8, 10 000 Zagreb, Croatia

**Keywords:** Epstein–Barr virus, latent membrane protein 1, signal transduction, cytokines, interferons

## Abstract

The disruption of antiviral sensors and the evasion of immune defences by various tactics are hallmarks of EBV infection. One of the EBV latent gene products, LMP1, was shown to induce the activation of signalling pathways, such as NF-κB, MAPK (JNK, ERK1/2, p38), JAK/STAT and PI3K/Akt, via three subdomains of its C-terminal domain, regulating the expression of several cytokines responsible for modulation of the immune response and therefore promoting viral persistence. The aim of this review is to summarise the current knowledge on the EBV-mediated induction of immunomodulatory molecules by the activation of signal transduction pathways with a particular focus on LMP1-mediated mechanisms. A more detailed understanding of the cytokine biology molecular landscape in EBV infections could contribute to the more complete understanding of diseases associated with this virus.

## 1. Introduction

Epstein–Barr virus (EBV), known otherwise as Human gammaherpesvirus 4, is a human double-stranded DNA virus belonging to the subfamily *Gammaherpesvirinae* and genus Lymphocryptovirus [1]. The virus was identified in 1964 in cultured lymphoblasts from Burkitt’s lymphoma [2]. Although it originated from patients from sub-Saharan Africa, it was soon apparent that EBV is a ubiquitous virus which latently infects more than 95% of the adults worldwide. As the spread of EBV is primarily mediated by saliva, most individuals are asymptomatically infected with the virus during early childhood [3]. An acute infection is mostly present among young adults as a causal agent of infectious mononucleosis (IM), a benign disease characterized by fever, lymphadenopathy and swollen liver or spleen. The virus establishes a lifelong persistency in its host, with possible reactivation in the case of a weakened immune system [4]. Following its discovery, EBV was found to infect resting human B lymphocytes and to induce their continuous proliferation into lymphoblastoid cell lines (LCLs) in vitro, which led to the virus being recognised as the first known human oncovirus [5,6]. To this day, EBV is classified as a group I carcinogen and its associated tumours account for 1.5% of all human cancers. Although the EBV genome in infected cells usually exists as an episome, its integration into a host genome proves to be a significant factor for promoting the development of diverse types of tumours [7]. Primarily, EBV is etiologically associated with malignancies of epithelial and lymphatic origin, such as nasopharyngeal carcinoma (NPC), Hodgkin Lymphoma (HL), gastric cancer (GC), extranodal NK/T cell lymphoma and post-transplant lymphoproliferative carcinoma (PTLD) [8].

Latent membrane protein 1 (LMP1) is considered to be the main EBV oncogene. Its potential to transform cells was first discovered by Wang et al. in rodent fibroblasts and it was later affirmed as an essential EBV protein for B cell transformation [9,10]. LMP1 can induce multiple cell growth and survival signals by mimicking the constitutively active B cell receptor CD40, which is a member of the tumour necrosis factor receptor (TNF-R) family [11]. Aside from a tumourigenic function, LMP1 also plays an extensive role in antiviral immune response, cytokine and chemokine induction, and immune evasion. Some of the pathways modulated by LMP1 include NF-κB, MAPK, JNK, JAK/STAT and PI3-K [12]. The molecular features of EBV infection are exceptionally complex with multiple viral proteins interacting with cellular signalling pathways that ultimately shape the biology of the infected cell. The focus of this review is to provide a molecular background for the role of EBV latent membrane protein 1 within the context of the cytokine network that characterizes the immune response to this virus.

### EBV Cell Entry

In order to successfully establish infection, EBV has developed different cell entry mechanisms depending on its two major targets—B lymphocytes and epithelial cells [13]. The well-studied entry into B lymphocytes begins with endocytosis and then proceeds with the fusion of viral and vesicular membranes, while the less characterized entry into epithelial cells mainly occurs by direct fusion of the virus with the cell plasma membrane [14]. The B cell surface receptors CD21 (complement receptor 2 (CR2)) and CD35 (complement receptor 1 (CR1)) are binding sites for the most abundant EBV glycoprotein, gp350. This initial attachment allows the viral glycoprotein gp42 to bind to the human leukocyte antigen class II (HLA-II) and activate glycoprotein gB, which is a conserved final executor of fusion in both host cell types [15,16]. gp42 is a unique EBV glycoprotein which functions as a tropism switch. It selectively infects B cells by coupling into a trimeric complex with gH/gL and blocking interaction with the epithelial cell surface receptor. However, gp42 is absent from newly synthesized virions from B cells since binding to HLA-II eventually leads to its degradation in the endoplasmic reticulum. Thus, those virions show tropism for epithelial cells [17,18].

Unlike in B cells, the infection of epithelial cells begins with the attachment of EBV glycoprotein BMRF2 to host surface integrins and their interaction with the heterodimeric gH/gL complex [19]. Additionally, it was recently found that the ephrin receptor tyrosine kinase A2 (EphA2) holds an important role as an entry receptor for EBV through binding to gH/gL independently of gp42 [20,21]. However, it needs to be emphasised that the efficient receptor function of EphA2 has only been confirmed in cancer cell lines, as its accessibility has not been shown in healthy gastric organoids [22]. Progress has also been made in the study of the EBV entry mechanism for T lymphocytes, as CD21, just like in B cells, was revealed to be the necessary surface receptor for EBV gp350 [23].

## 2. Innate Immune Response to Epstein–Barr Virus

Activation of the host innate immune response is an immediate antiviral reaction to EBV infection, which is characterized by the generation and release of various pro-inflammatory cytokines like chemokines, interleukins (ILs), interferons (IFNs) and tumour necrosis factors (TNFs) [24]. An infected cell detects viral invasion through binding viral components containing pathogen-associated molecular patterns (PAMPs) with its pattern recognition receptors (PRRs). PRRs include various membrane and cytosolic sensory proteins such as Toll-like receptors (TLRs), RIG-1-like receptors (RLRs) and NOD-like receptors (NLRs) [25]. Not only does the PRRs’ recognition activate one of the essential antiviral defences, a type I IFN response, but also other diverse immune signalizations such as inflammasomes, the NF-κB pathway and apoptosis [25,26]. Nevertheless, viruses like EBV have evolved quite a few countermeasure strategies to evade those immune defences. They usually aim to disrupt antiviral sensors and utilize components of signalling cascades to their needs, that is, to promote viral replication and persistence (Figure 1). So, it is their interplay that decides the final outcome of the EBV infection [24,26].

### 2.1. Inflammasome Impact on EBV Infection

In response to EBV infection, sensory proteins from the TLR/NLR family trigger the assembly of intracellular multiprotein complexes known as inflammasomes. Their role is to incite pro-inflammatory caspases, like caspase-1, which in return release cytokines IL-1β and IL-18 and lead to pyroptosis [27]. A two-step inflammasome activating process begins with priming signal 1, which stimulates TLRs and cytokine receptors (TNF or IL-1 receptors) for induction of the transcription of pro-IL-1β and NLRP3 through NF-κB activation. After that, signal 2 directly activates sensory proteins depending on each inflammasome type, such as NLRP3 inflammasome in the case of PAMP recognition [28].

As one of the main triggered cytokines in early inflammation, IL-1 is an invaluable activator of host immune antiviral response. But it is precisely that activation role that can prove detrimental to the host. In the case of aberrant cytokine regulation, it can lead to unrestrained inflammation and tissue damage, thus consequently providing a chance for infected cells to survive [29]. EBV utilizes that inflammasome flaw to enter into the lytic phase. Through the exploitation of several inflammasome sensors, EBV influences caspase-1 mediated depletion of the epigenetic repressor and zinc finger protein KAP1. KAP1 is an EBV lytic gene blocker which enforces gene heterochromatinization, particularly of gene BZLF1. Therefore, the derepression of the BZLF1-encoded protein ZEBRA frees the vital EBV replication transactivator [30]. In addition to that, EBV can also use EBERs to increase the expression of cytokines IL-1α, IL-6 and TNFα (via the TLR3 pathway) for cancer-promoting inflammatory response in vitro and in vivo [31]. In addition, since IL-1 and IL-18 signalling can induce an apoptosis-promoting type I IFN response, prevention of their receptor binding by their sequestration in the exosomes can further facilitate EBV persistence [29].

Autophagy, a crucial process for intracellular recycling and the elimination of infectious particles, is also regularly initiated during the innate immune response [32]. But, if the antiviral response becomes excessive, inflammatory signals can in turn activate autophagy to suppress it and protect the host. This balancing role is quite important in defence against some inflammatory diseases associated with NLRP3 inflammasomes [33]. Autophagy downregulates the NLRP3 inflammasome by several mechanisms; some of them are the degradation of its activators, like cytokine IL-1β, or the utilization of the p62-dependent sequestration of its components. [34]. The EBV latent protein LMP1 has been proposed to be a dose-dependent activator of autophagy [35]. While low levels of LMP1 in cells stimulate early stages of autophagy, namely autophagosomes, highly expressed LMP1 instead causes the formation of late-stage autolysosomes. Furthermore, cells with high levels of LMP1 show dependence on the degradation of LMP1 via autophagy. That degradation also seems to control the cytostatic effect of LMP1 as it is a prerequisite for the proliferation of B cells. Overall, it seems that the goal of this LMP1 regulation is to limit its own accumulation to provide EBV-infected cells a direct escape from the host immune recognition. [35,36]. But, the exact mechanism that LMP1 uses to subvert autophagy is still inconclusive. In contrast to the initial study by Lee and Sugden (2008), a more recent study on EBV-positive B cells points to the involvement of the NF-κB pathway [36].

### 2.2. Modulation of Type I IFN Response

The activation of Toll-like receptors, RIG-1-like receptors and other PRRs is a prerequisite for the secretion of type I interferons during the immune antiviral response [37]. Plasmacytoid dendritic cells (pDCs) are the major source of massive IFN-I production and are responsible for the constitutive expression of IFN-I activators TLR7, TLR9 and interferon regulatory factors (IRFs). EBV has developed diverse mechanisms to avoid nearly all segments of the IFN-I signalling cascade [38]. During the primary infection, multiple lytic EBV proteins and EBERs are involved in the downregulation of TLR7 and TLR9 [39]. As other detectors of viral RNAs in the cytoplasm, RIG-1-like receptors are also important targets for inhibition by BART miRNAs, like miR-BART6-3p [40]. Moreover, the essential IFN transcription factors IRF3 and IRF7 are usually inhibited by diverse viral lytic proteins, such as BGLF4 and BFRF, that interact with IRF3, or BZLF1, that interacts with IRF7 [Wang et al. 2020]. Contrary to the lytic cycle inhibition, these two IRFs were found to be induced in EBV-facilitated transformation during the latent viral phase, especially IRF7, which is activated by EBV main oncoprotein LMP1 [41].

Other than the already mentioned role in the immunomodulation of inflammasomes, autophagy also has an impact on IFN-I signalling. Autophagosomes can deliver and present PAMPs to both TLR and RIG-1 receptors in order to initiate an IFN-I response. Correspondingly, IFN-I and ISG products also regulate autophagic processes [42,43]. This cross-talk is known to be targeted by EBV [42]. A study on the pDCs implied that EBV impaired their maturation via the induction of TLR9 and TLR7 located on lysosomes [44]. So, it all leads to a proposition that autophagy may have a dual function. On one side, it can use antiviral activity for the elimination of viruses, and on the other side, it can use a proviral function to help with viral replication or cell exit [44,45]. On top of that, this interplay insinuates the significance of EBV protein LMP1. Since LMP1 is known to be a downregulator of TLR9, RIG-I and the downstream IFN-I cascade, as well as an autophagy modulator, it should be interesting to uncover the deep mechanisms influencing this interplay [35,46,47,48].

Binding of the secreted type I IFNs to their specific IFN-α/β receptor 1 and 2 (IFNAR1 and 2) activates the JAK/STAT signalling pathway. This pathway is formed of two conserved components: the receptor Janus tyrosine kinase (JAK) and the signal transducer and activator of transcription (STAT) [49]. Heterodimers of IFNAR1 and IFNAR2 phosphorylate tyrosine kinases JAK1 and TYK2, which in turn phosphorylate STAT1 and STAT2 to form heterodimers. The STAT heterodimer then proceeds to the nucleus to bind with IRF9 and consequently form a transcriptional activator known as an IFN-stimulated gene factor (ISGF). ISGF integrates with a transcriptional enhancer known as an IFN-stimulated response element (ISRE), which holds the role of the final cascade activator of IFN-stimulated genes (ISGs) [50,51].

Components of the JAK/STAT cascade are one of the main antagonists for EBV immune evasion. For the start, EBV oncoproteins LMP-2A/2B disrupt the membrane receptors, IFNARs, by increasing their degradation and thus break down the initial IFN binding [52]. The next known step involves the modulation of STAT1, which actually represents one of the most frequent EBV targets in IFN signalization. STAT1 has a dual role in IFN type I signalization; on one side, it is an immune surveillant of infected host cells, but on the other side it is a maintainer of viral latency [53,54]. For instance, EBV tegument protein BGLF2 inhibits STAT phosphorylation through linkage with TYK2 and promotes the ubiquitination and proteasomal degradation of STAT2. These interactions not only downregulate IFN signalization but also oppose the repressive ability of IFN- α/β in the case of EBV reactivation [54,55]. In contrast, latent EBV proteins like EBNA1 and LMP1 enhance STAT1 phosphorylation [56,57]. In addition to STAT1 and 2, another member of that family plays a role in cytokine signalization. STAT3, which is in fact activated ahead through the IFN-I response, in return modulates cytokine activity by downregulating the ISG expression induced via IFN-α [58]. That negative regulation is naturally exploited by viruses like EBV. EBNA2 and LMP1 elicit STAT3 activity so that they can engage in the inhibition of EBV lytic replication and participate in B cell proliferation [59,60]. Apart from EBNA2 and LMP1, some EBV miRNAs like miR-BART1 and miR-BART16 also contribute to the suppression of ISGs [39].

The regulation of IFN secretion additionally depends on one other negative feedback mechanism that involves the induction of suppressor of cytokine signalling (SOCS) proteins. SOCS1 and SOCS3 are directly influenced by STAT proteins activated in the IFN-α/β pathway [61]. SOCS3-mediated ubiquitylation of the kinase activity of JAKs and cytokine receptors induces ISGs to monitor IFN signal propagation and antiviral inflammation [62]. For the successful downregulation of SOCS1, EBV employs LMP1 or transcription factor EBNA2 to enhance the expression of host cellular miR-155. This oncogenic miRNA greatly contributes to the B-lymphoproliferation and survival of transformed B cells in plenty of lymphomas [63].

## 3. Specific Immune Response to EBV

The antiviral innate immune response sets in motion all the main components of the adaptive immune response: the cytotoxic CD8+ T cells, CD4+ T helper cells and antibody-producing B cells. While CD8+ cells have the ability to directly attack and get rid of the infected cells, CD4+ cells mediate immune responses through the secretion of specific cytokines [64]. This response is exemplary in acute IM, which has a characteristically exaggerated expansion of CD8+ T cells that are mostly specific for EBV lytic phase antigens and can account for up to almost 50% of all blood circulating lymphocytes [65,66].

Effector CD8+ T cells recognize infected cells through a specific T cell receptor (TcR) that binds to antigen-presenting HLA class I (HLA I) molecules and release the perforin and granzyme proteins [67]. Since all nucleated human cells express HLA I, it is no wonder that EBV has developed more than one independent strategy for HLA I downregulation and interference with CD8+ T cell surveillance. The three main EBV proteins that enable this evasion are early lytic cycle proteins, BGLF5, BNLF2a and BILF1 [68]. The BGLF5 protein interferes with T cell recognition via the promotion of HLA I-encoded mRNA degradation, but it is not potent enough to completely impair it, unlike the other two proteins, BNLF2a and BILF1 [69]. For instance, BNLF2a is a vital EBV inhibitor of the transporter associated with antigen processing (TAP) protein complex. It can shut down peptide/HLA presentation through the blockage of peptides and ATP binding to the TAP complex [70]. The third evasion molecule is an EBV-encoding gene, BILF1, which encodes a constitutively active G-protein-coupled receptor (GPCR) that not only downregulates HLA I via endocytic and exocytic pathways, but also induces signalling-mediated tumourigenesis [71].

Although most of the EBV mechanisms to prevent the expression of MHC antigens involve lytic gene products, latent cycle proteins also play an important part. A well-known B cell receptor mimic, LMP2A, can indirectly supress MHC class II expression via the downregulation of its master regulator, CIITA, and other B cell transcription factors [72]. However, in case of LMP1, MHC modulation becomes a bit contradictory. Even though LMP1 greatly elicits MHC-I presentation via activation of the NF-κB pathway, T cell responses to LMP1 in healthy carriers are rarely detected [73,74]. This is due to the ability of LMP1 to limit its self-presentation to CD8+ T cell epitopes. The exact mechanism is still a bit elusive, but it is known that LMP1 constrains its *cis*-acting elements through aggregation, which probably renders LMP1-derived T cell epitopes vulnerable to degradation by cellular proteasomes [74].

### CD8+ T Cell Response in Chronic EBV Infection

Chronic viral infection and cancer lead to a prolonged exposure of the host immunity to high antigen levels and, thus, progressive loss of T cell responses, which can result in T cell exhaustion [75]. Exhausted T cells decrease production of major pro-inflammatory cytokines in a hierarchical manner, starting from IL-2 and TNF to IFN-γ and eventually death [76]. IL-2 is one of the main drivers for CD8+ T cell exhaustion, as it is a T cell growth factor that is crucial for their proliferation and persistence. Additionally, it is also involved in the production of effector and memory cells [77]. Adversely, this mechanism boosts the expression of inhibitory cytokines like IL-10 and TGF-β and receptors like programmed cell death protein 1 (PD-1), as well as provides the constitutive expression of inflammatory chemokines like CCL3 and CCL5 [78,79]. Chemokine CCL5 recruits various immune cells, including T cells, in response to infection or injury. Therefore, during prolonged antigen exposure, it facilitates the formation of an immune-suppressive environment [80].

In the case of persistent EBV infection, the latent phase protein EBNA3 provokes the highest immunogenicity for CD8+ T cells, and other EBV proteins, like EBNA1, LMP1 and LMP2, have also been identified as its targets [81,82]. So, to efficiently evade T cells and establish latent persistence, EBV employs a wide range of regulatory mechanisms. For example, one of the noteworthy ones specifically targets immunomodulatory cytokine IL-10. During the initial infection, EBV uses the viral IL-10 (vIL-10) encoded by the lytic gene BCRF to mimic human IL-10 and suppress the secretion of IL-2 and IFN-γ [83]. Furthermore, during EBV latency, LMP1 and LMP2A increase the production of cellular IL-10 [84,85]. This coordinated action of both vIL-10 and cIL-10 is necessary for the viral control of specific immunity, survival and expansion of EBV-infected naive B cells. While it has been shown that vIL-10 can repress TAP1 and HLA I, cIL-10 can directly reduce the TCR signalling capacity and, thus, the antigen sensitivity of CD8+ T cells [86,87,88]. Moreover, it has to be mentioned that EBV miRNAs also play an important role in adaptive immune silencing. Apart from downregulating the TAP complex, miRNAs can also decrease the generation of a powerful T cell-simulating cytokine, IL-12, and chemoattractant, CXCL11 [89].

In addition to the modulatory effect on IL-10, two latent EBV gene products, LMP1 and LMP2A, can employ some other measures to bypass the specific immune response. For instance, Ouyang et al. showed that, in LCLs and EBV+ PTLD tumours, both gene products indirectly promoted apoptosis of EBV-specific CD8+ and CD4+ T cells through the transcriptional activation of immunomodulator Galectin-1 (Gal1) [90]. A recent study unveiled that LMP1 can also elevate the levels of extracellular vesicles expressing PD-L1. PD-L1 is a ligand which, via binding to CD8+ T cells, inhibits T cell regulation and the production of interleukins [91]. As well as LMP1, LMP2A can also interfere with CD8+ T cell recognition of EBV-infected B cells. One of the inhibitory mechanisms discovered on LCLs includes impairment of the activation of NKG2D, an agonistic T and NK cell receptor on the surface of CD8+ T cells [92]. On top of that, LMP2A is also implicated in the reduction in the expression level of EBV latent antigens targeted by CTLs, though this antigen-specific regulation is still partially unknown and conflicting [92,93,94].

## 4. EBV-Mediated Induction of Immunomodulatory Molecules by Activation of Signal Transduction Pathways

Today, several signalling pathways included in chemokine and cytokine production are known to be triggered by EBV-encoded LMP1, a 60–66 kD integral transmembrane protein which consists of 386 amino acid (aa) residues. The LMP1 molecule comprises a short N-terminal cytoplasmic tail (aa 1–24), six transmembrane domains (aa 25–186) and a long C-terminal cytoplasmic tail (aa 187–386) (Figure 2) [95,96]. The amino-terminal segment is primarily responsible for correctly orientating LMP1 to the plasma membrane and serves as the attachment site for the ubiquitin pathway [97,98]. The transmembrane segment promotes the intermolecular oligomerization and aggregation of LMP1 molecules as well as actin cytoskeleton rearrangement via interaction with Cdc42, which is a member of the Rho (Ras-homology) GTPase family [95,99]. However, the majority of LMP1’s impact on cellular homeostasis is achieved via the C-terminal domain and its three functional subdomains or C-terminal activating regions (CTAR1, CTAR2, CTAR3). The CTAR1 domain contains the PxQxT motif (aa 204–208), which is necessary for the interaction of LMP1 with tumour necrosis factor receptor-associated factors (TRAFs); the last three amino acids in the CTAR2 region form the YYD motif (aa 384–386), which is important for interaction with the tumour necrosis factor receptor type 1-associated death domain protein (TRADD) and BS69, a multi-domain-containing cellular protein, while the CTAR3 region (aa 275–330) is a proline-rich region located between CTAR1 and CTAR2 which was shown to successfully bind Janus kinase 3 (JAK3) [11,57,100,101,102,103].

Even though the substantial effect of the LMP1 protein on cell activation was observed previously, its precise signal transduction pathways and their biochemical mechanisms were unknown [104]. One of the first reports of the LMP1-mediated induction of the NF-κB pathway was demonstrated by Laherty et al. (1992), who showed the LMP1-induced expression of the A20 zinc finger protein by binding an NF-κB-like factor to κB sites within the A20 promoter [105]. However, this mechanism alone was not sufficient to explain the observed impact of LMP on cell transformation and immortalization. In the following years, Kieser et al. (1997) identified the c-Jun N-terminal kinase (JNK) pathway as one of the effectors of LMP1 by induction of the activity of the AP-1 transcription factor [106], while Roberts and Cooper (1998) demonstrated that LMP1 activates the extracellular signal-regulated kinases 1/2 (ERK 1/2) pathway shown to be necessary for the malignant transformation of rat fibroblasts [107]. Furthermore, Elioupoulos et al. (1999) demonstrated that LMP1 also engages the p38 mitogen-activated kinase pathway through both the CTAR1 and CTAR2 domains mediated by the adaptor protein TRAF2, independent of NF-κB activation [108]. Gires et al. (1999) identified a third C-terminal activating region between CTAR1 and CTAR2 of LMP1 which is required for the activation of JAK3 and the Janus kinase/signal transducer and activator of transcription proteins (JAK/STAT) signalling pathway [11]. Finally, Dawson et al. (2003) showed that LMP1 can activate the phosphatidylinositol 3-kinase/protein kinase B (PI3K/Akt) pathway by association with the p85 regulatory subunit of PI3K with a profound effect on cell survival and proliferation [109]. We briefly discuss the role of LMP1 in each signalling pathway and the recent data on the LMP1-mediated induction of cytokines and chemokines as crucial inflammatory mediators (Figure 2). It should also be emphasised that the induction of signal transduction pathways and immunomodulatory molecules by LMP1 may vary according to the cell type used in a particular study.

### 4.1. LMP1 and NF-κB Pathway Activation

Lai et al. (2010) showed that LMP1 activated both canonical and non-canonical NF-κB pathways in NPC cell lines by using LMP1 mutants to determine the necessity of various CTARs for particular signalling pathway variants [110]. In the canonical NF-κB pathway, the recruitment of adaptors (TRAFs and death domain kinase RIP) after ligand binding to a variety of cell receptors recruits and activates the enzyme IκB kinase (IKK) complex which phosphorylates members of the inhibitor of kB (IκB) family, such as IkBα, leading to its ubiquitination and degradation, which enables canonical NF-κB family members, such as NF-κB dimer p50-RelA, to enter the nucleus and induce the transcription of target genes [111,112,113]. On the other hand, the non-canonical pathway is based on the processing of p100 and is typically activated by the binding of ligands, such as members of the tumour necrosis factor (TNF) receptor superfamily, who initiate the activation of NF-κB-induced NIK kinase which leads to p100 phosphorylation via the activated kinase IKKa complex; the liberation of non-canonical NF-κB family members, such as NF-κB dimer p52-RelB; and their nuclear translocation [111,112,114]. Lai et al. (2010) showed that the CTAR1 and CTAR2 of LMP1 are required for canonical and non-canonical NF-κB pathway activation, respectively [110].

NF-κB signalling activated by LMP1 was shown to induce the production of various immunomodulatory molecules. One of the first studies by Eliopolous et al. (1997) demonstrated the induction of IL-6 production via NF-κB pathway activation in SV40-transformed keratinocytes and human carcinoma cell lines [100]. IL-6, a pleiotropic cytokine, was shown to have a crucial role in inflammation, with recent data suggesting it is also one of the most important cytokines during viral infection, due to several potential mechanisms which can induce viral persistence, immune evasion and ultimately lead to chronic infection [115]. Furthermore, Ren et al. (2004) showed in epithelial/NPC hybrid cell line NPC-KT that LMP1 induces IL-8 by activation of NF-κB via the NF-κB binding site in the IL-8 promoter [116], while Vockerodt et al. (2005) demonstrated that the LMP1-mediated expression of IP-10 in lymphoma cells is independent of inducers such as IFN-γ, IL-18 and TNF-α and its promoter activity can be regulated by NF-κB signalling along with the p38 pathway [117]. Therefore, the ability of LMP1 to act as a strong chemokine inducer further increases the inflammatory environment by the upregulation of infiltrating immune cells and exacerbating oncogenic processes [118]. LMP1-induced NF-κB signalling was shown to play a key role in IL-8 production in a human gingival epithelial cell line [119]. IL-1 induction by LMP1 was observed by Huang et al. (2010), who detected the upregulation of IL-1α and IL-1β secretion in epithelial cells [120]. LMP1-mediated induction of the NF-κB pathway may result in the activation of IL-1α and IL-1β promotors, and the ample production of IL-1α and IL-1β may facilitate lymphocyte infiltration of an NPC tumour [120]. Interestingly, it was observed that LMP1-mediated NF-κB activation has many similarities with IL-1β induced NF-κB signalling, with 118 out of the 155 LMP1 NF-κB activation pathway components being similarly important for IL-1β-induced NF-κB signalling in HEK293 cells [121]. A recent report showed that IL-1β may serve as the inducer of angiopathy in systemic chronic active Epstein–Barr virus infection [122]. Furthermore, LMP1-mediated NF-κB activation was shown to induce survival signals not only in B cell lines but also in T and NK cells, suggesting its potential role in the development of EBV-positive T or NK cell neoplasms [123].

### 4.2. LMP1 and MAPK Pathway Activation

MAP (mitogen-activated protein) kinase pathways share a common sequentially activated cascade which begins with the activation of MAPKKK (MAPK kinase kinase) via cell surface receptors and the subsequent phosphorylation and activation of MAPKK (MAPK kinase), which in turn activates a particular protein, Ser/Thr kinase MAPK, by dual phosphorylation of the conserved Thr-X-Tyr motif, leading to the regulation of various crucial processes such as cell growth, development, differentiation, survival, apoptosis and death [124,125]. Three main families of mammalian conventional MAP kinases are extracellular signal-regulated kinases (ERKs), c-Jun N-terminal kinases (JNKs) and p38 mitogen-activated protein kinases (p38s) [125]. LMP1-mediated induction was observed for all three MAPK pathways.

#### 4.2.1. LMP1 and JNK Pathway Activation

JNKs, which are also called stress-activated kinases (SAPKs), were initially implicated in response to stress stimuli, but were later found to respond to various cytokines and growth factors. After the stimulation of MAPKKKs, the phosphorylation and activation of MAPKKs MKK4 (JNKK1) and MKK7 (JNKK2) in turn activates MAPK JNK which phosphorylates c-Jun and other members of AP-1 transcription factors [126,127]. LMP1-induced activation of the JNK pathway was first found to be CTAR2-mediated and lead to the induction of transcription factor AP-1 activity, thus impacting the proliferation, transformation and immortalization properties of different cell types [106,108]. The ability of CTAR2 to interact with the N-terminus of TRADD and the later recruitment of proteins such as TRAF2 at first seemed to have a central role in LMP1-mediated JNK pathway signalling [108]. Further research showed that JNK pathway activation may also be inducted by the CTAR1 domain of LMP1, but only in the presence of TRAF1 [112]. However, LMP1 CTAR1-induced JNK activation was shown to be indirect and most likely occur through the TRAF2 protein [128]. TRDD was first discovered as a mediator of programmed cell death signalling and activation of NF-κB by its ability to both interact with the death domain (DD) of TNFR1 and recruit effectors such as TRAF2 and the FAS-associated death domain protein (FADD) [129,130]. It seems that the LMP1-mediated association of TRADD differs from the interaction between TNFR1 and TRADD both structurally and functionally since LMP1 does not have a death domain (DD) and favours the binding of TRAF2 instead of FADD, therefore only promoting anti-apoptotic signalling and the upregulation of genes with binding sites for the AP-1 transcription factor [103,131,132]. Further research showed that the sequential activation of MAPKKK TAK1 (transforming growth factor β (TGF-β) activated kinase 1), TAB1 (TAK1 binding protein), TRAF6 (which serves to bridge LMP1 and TAB1) and MAPKKs JNKK1 and JNKK2 as activators of MAPK JNK are essential in the LMP1-mediated JNK pathway [133]. Recently, the unexpected role of the IKK isoform IKK2, which is otherwise essential for LMP1-induced NF-κB signalling, was shown in JNK activation by Voigt et al. (2020). The absence of IKK2 almost completely blocked LMP1-induced JNK activation, while serine/threonine kinase TPK2 or MAP3K8 was shown to transmit JNK activation signals downstream of IKK2, with TAK1 kinase activity playing a role in JNK1 activation concurrently with IKK2 (Voigt et al. 2020). While this cross-talk between NF-κB and JNK signalling modules may have important implications in EBV-induced cellular transformation, it also provides potential therapeutic targets [134].

One of the first studies showing the impact of LMP1-induced AP-1 transcription factor activation observed a positive regulatory role of LMP1 in IL-8 synthesis [108]. LMP1-mediated phosphorylation of c-Jun through the activation of JNK seems to be important in the modulation of IL-8 expression and production [108]. Lai et al. (2010) showed the LMP1-induced production of macrophage inflammatory protein-1 alpha (MIP-1α/CCL3) and macrophage inflammatory protein-1 beta (MIP-1β/CCL4) in EBV-infected NPC cell lines requiring both NF-κB and JNK signalling pathways [110]. Treatment with either IB kinase inhibitor X, which hindered both canonical and non-canonical NF-κB pathways, or with a JNK inhibitor, SP600125, which inhibited JNK phosphorylation/activation, prevented LMP1-triggered production of the two chemokines [110]. MIP-1α and MIP-1β are linked to leukocyte recruitment in response to tissue injury or inflammation and the regulation of acute and chronic inflammatory host responses [135,136]. The induction of various chemotactic molecules by LMP1 may contribute to lymphocytic infiltration and tumour development.

#### 4.2.2. LMP1 and ERK1/2 Pathway Activation

Several studies initially observed LMP1-induced activation of the ERK-MAPK pathway to be necessary for the malignant transformation of rodent fibroblasts [107,137]. The mutation or aberrant expression of components of this signalling pathway, which is known to play a central role in the control of cell proliferation, often plays a major role in the tumourigenesis of human cancers [124,138]. The impact of ERK-MAPK pathway activation in LMP1-induced human epithelial cells was identified in association with cell motility, suggesting its contribution to LMP1 oncogenicity [139]. The canonical ERK-MAPK cascade includes activation of the Ras protein after the binding of the ligand to the receptor and its stimulation of the MAPK kinase kinase (MAPKKK) Raf, MAPK kinase (MAPKK) MEK1/2 and MAP kinase (MAPK) ERK1/2, leading to the transcription of genes involved in several cellular control mechanisms [140]. The activation of ERK1/2 mediated by MEK1/2 was mapped to CTAR1 of LMP1, with important factors being TRAF2 and TRAF3 [137,139]. LMP1-induced ERK-MAPK pathway activation was therefore observed to occur in an Ras-independent manner [139]. Several mechanisms have been proposed, such as the activation of MAPKKK c-Raf or the direct phosphorylation of MEK1/2 by LMP1, leading to the downstream activation of p90RSK (ribosomal s6 kinase), also known as MAPK-activated protein kinase-1 (MAPKAP-K1), and the regulation of transcription factor c-Fos [139,141].

Constitutive ERK-MAPK phosphorylation induced by LMP1 was observed to obstruct TGF-β mediated growth inhibition via the TGF-β1/ERK-MAPK/p21 pathway, therefore deregulating cellular growth and creating a permissive environment for tumour development [142]. TGF-β is a pleiotropic cytokine essential for immune regulatory mechanisms which may serve as a tumour suppressor in the early stages of tumour development [143,144]. Even though SMAD-dependent TGF-beta signalling (which is out of the scope of this review) was shown to be the main driver of epithelial-to-mesenchymal transition (EMT), cell adhesion and spreading, several recent studies confirmed an important role of the LMP1-induced ERK-MAPK pathway in cell motility, migration, the enhancement of invasion properties and many of the features of the EMT [145,146,147,148]. Therefore, the cross-talk between ERK-MAPK and TGF-β signalling suggests that LMP1 may utilize multiple signalling pathways to elicit specific cellular responses [145]. Furthermore, the activation of ERK-MAPK by LMP1 via the CTAR1 domain was shown to be required for the inhibition of AMPK (AMP-activated protein kinase) which is a crucial regulator of intracellular energy balance, therefore contributing to LMP1-mediated proliferation [149].

#### 4.2.3. LMP1 and p38 Mitogen-Activated Kinase Pathway

The activation of p38 kinases is initiated by dual phosphorylation by MAPKK kinases MKK3 and MKK6 in response to various extracellular stimuli and a diverse range of MAPKKK kinases [150]. Environmental stress signals and inflammatory cytokines, similar to the JNK module, seem to be the main driver of p38 pathway activation, which plays an important role in maintaining homeostasis, immune response regulation, cell proliferation and survival [124,151]. The discovery of p38 MAPK pathway engagement by LMP1 was attributed to both the CTAR1 and CTAR2 domains and was found to be mediated by the TRAF2 protein [108]. Other signalling molecules found to transmit LMP1 signals to p38 MAPK downstream of TRAF2 were found to be TRAF6 and MAPKK MKK6 [152]. Furthermore, Schultheiss et al. found the PxQxT motif of CTAR1 and tyrosine 384 of CTAR2 necessary for the full signalling capacity of LMP1 to p38 MAPK [152]. Not only is the p38 signalling pathway itself activated by LMP1, but the increase in p38 pathway activation was also shown to upregulate LMP1 expression, creating a positive autoregulatory loop [153].

The first immunomodulatory molecules found to be induced by LMP1 via the p38-regulated mechanism were found to be IL-6 and IL-8, which are key mediators associated with various inflammatory pathways. LMP1′s capacity to promote the expression of IL-6 and IL-8 was discovered to be impaired by inhibiting p38 signalling [108]. Induction of IL-10 by LMP1 was observed in Burkitt’s lymphoma suggesting a necessity of both CTAR1 and CTAR2 and the potential involvement of the p38 pathway [154]. IL-10, known as a cytokine synthesis inhibiting factor with strong anti-inflammatory properties, is tightly regulated with low expression levels in nonpathological conditions [155,156]. On the other hand, the induction of IL-10 was previously known to be associated with the proliferation of EBV-transformed B cells and lymphomas [157]. Lambert et al. (2007) showed that CTAR2 is predominantly required for p38 activation by LMP1 and that downstream p38-dependent signalling contributed to CREB transcription factor activation, which is known to participate in IL-10 production [84]. Furthermore, an autocrine regulatory loop between IL-10 and LMP1 was observed with IL-10 inducing the expression of LMP1 [158]. Protein kinase PKR, which was demonstrated to be involved in activating p38 MAPK phosphorylation, was found to be an upstream mediator of LMP1-induced IL-6 and IL-10 upregulation [159]. More recent data showed that, besides EBV plasma viral load, the ratio of LMP1-induced IL-8 and IL-10 may have prognostic value in NPC as a marker of disease progression [160]. Furthermore, the presence of LMP1 in exosomes was shown to increase the oncogenic properties of NPC and promote radioresistance by p38 MAPK signalling activation in surrounding cells [161].

### 4.3. LMP1 and JAK/STAT Signalling Pathway Activation

As previously described, the JAK/STAT pathway is one of the most straightforward signal transduction pathways allowing the rapid impact of extracellular signals on the regulation of gene expression [49,162]. It should be emphasised that the nature of this pathway is highly versatile—while there are four members of the JAK family (JAK1, JAK2, JAK3 and TYK2), which phosphorylate tyrosine residues of the receptors, therefore creating binding sites for members of the STAT family (STAT1, STAT2, STAT3, STAT4, STAT5a, STAT5b and STAT6) which are responsible for the induction of target gene transcription after translocation to the nucleus, activation of the specific JAK family members is ligand-dependent [163,164]. The identification of the JAK/STAT pathway as one of the targets of LMP1 was first based on the observed interaction of JAK3 and the proline-rich region with the 33pb repetitive sequence of LMP1 C-terminus (CTAR3) which upregulated STAT1 and STAT3 DNA binding activity [11]. Further research into the molecular mechanisms of STAT3 phosphorylation induced by LMP1 supported the involvement of the JAK/STAT signalling pathway in LMP1-regulated STAT3 activation [165,166].

Klein et al. (1995) first showed the expression of IL6 and IL6 receptor (IL6R) in Burkitt’s lymphoma and the potential involvement of IL/IL6R in the growth and differentiation of EBV-infected B cells [167]. IL-6 is a pleiotropic 25 kDA protein with diverse roles in the regulation of the immune system and metabolism and, consequentially, the pathophysiology of many diseases [168,169]. Upregulation of IL-6 expression by LMP1 was previously demonstrated by other signalling pathways, such as NF-κB and p38 MAPK [100,108]. On the other hand, IL-6 has also been known to activate the JAK/STAT signalling pathway through STAT3 [170]. After the identification of the JAK/STAT pathway as the target of LMP1, studies have found that the LMP1-mediated increased expression of IL-6 could instigate a positive feedback loop of continuous STAT3 activation independent of the initial stimulus [165,171]. Recent research into the influence of human anti-LMP1 IgG antibody (LMP1-IgG) on the JAK/STAT signalling pathway on extranodal nasal-type natural killer (NK)/T cell lymphoma showed LMP1-IgG significantly inhibited the phosphorylation of STAT3 and JAK3, suggesting a potential therapeutic impact of targeting the JAK/STAT pathway [172]. Furthermore, the possibility of the successful treatment of EBV-associated lymphoproliferative disorders by targeting the JAK/STAT pathway was observed in mouse tumour models with an analog of the JAK2 inhibitor zhankuic acid A, antcin H, which demonstrated the suppression of cell proliferation [173].

### 4.4. LMP1 and PI3K/Akt Pathway Activation

The PI3K/Akt pathway is generally initiated by the stimulation of cell surface receptors, leading to the activation of PI3K which then phosphorylates phosphatidylinositol (3,4)-bisphosphate (PIP2) lipids to phosphatidylinositol (3,4,5)-trisphosphate (PIP3). The binding of Akt, also known as protein kinase B (PKB), to PIP3 and the subsequent modification and activation of Akt enables its translocation to the cytosol and nucleus [174]. Akt serves as a mediator of a variety of important cellular processes, such as cell growth, proliferation, repair, apoptosis and migration [175]. Dawson et al. (2003) first demonstrated that LMP1 interacts with the regulatory subunit of PI3K, p85, and that expression of LMP1 constitutively activates the PI3K/Akt pathway [109]. The CTAR1 domain of LMP1 was identified as the domain required for PI3K activation; however, no precise mechanism was elucidated, with TRAF molecules recruited to the CTAR1 domain being the most probable mediators [109,137]. Chen et al. (2008) demonstrated the requirement of the PI3K/Akt pathway for LMP1-mediated repression of DNA repair by the inactivation of FOXO3a (Forkhead box class O 3a) and the decrease of DDB1 (DNA damage-binding protein 1), which may lead to DNA damage accumulation and genomic instability [176]. This may contribute to the establishment of persistent EBV infection and potentially facilitate tumourigenesis [109,176]. Furthermore, a study by Meckes et al. (2010) showed that the transfer of LMP1 through exosomes to endothelial cells can activate the PI3K/Akt and ERK pathways, suggesting the potential modification of the tumour microenvironment by EBV via modulation of exosome components and functions leading to further tumour progression [177].

In addition to the impact on the genome integrity, LMP1 was shown to utilize the PI3K/Akt pathway for the induction of ectopic CD137 expression in Hodgkin Reed–Sternberg (HRS) cells, the malignant cells in Hodgkin Lymphoma [178]. CD137 is a member of the tumour necrosis receptor superfamily which was first observed to have a costimulatory role on activated T cells, activated NK cells and mature dendritic cells [179]. The underlying mechanism of PI3K/Akt pathway activation was shown to include mTOR and pS6K [178]. Furthermore, a study by Rajendran et al. (2016) demonstrated that ectopic CD137 expression on HRS cells enables the secretion of IL-13, facilitating immune escape by reducing IFNγ [180]. Decreases in IFN-*γ* secretion by cytokoxic T lymphocytes were observed in a recent study by Zhang et al. (2023) in children with infectious mononucleosis [181]. On the other hand, Lambert et al. 2007 demonstrated for the first time that the induction of IL-10 by LMP1 in B cells is not only p38-dependent but also uses the PI3K/Akt pathway [84]. Translational mechanisms, such as the PI3K/mTOR-dependent phosphorylation of p70-S6K, and the regulation of GSK3β, a kinase with various functions, have been one of the proposed mechanisms of IL-10 induction by PI3K. Even though no direct phosphorylation of transcription factor CREB was observed via the PI3K/Akt pathway, secondary effects on CREB were found to be mediated by GSK3β, therefore enhancing LMP1-mediated IL-10 production [84]. It should be emphasised that the IL-10 promoter contains a large number of transcription factor binding sites and the potential activation of other transcription factors by various LMP1-induced signalling pathways may synergistically impact IL-10 production [84,182].

## 5. Conclusions

A deeper understanding of the exact mechanisms and thorough analysis of immunomodulators induced by other EBV proteins could be of great importance in elucidating strategies of immunosuppression, and, ultimately, EBV-induced carcinogenesis. Furthermore, it provides insight into potential therapeutic targets for various EBV-associated disease pathologies. LMP1-mediated oncogenic pathways such as JAK/STAT, MAPKs/AP1 and NF-κB were shown to be at least partly responsible for the induction of PD-L1 expression in human NPC cells, enabling better efficacy of programmed cell death protein PD-1/PD-L1-directed immunotherapeutics [183]. The suppression of cell proliferation in LMP1-expressed lymphoma cells by JAK2 inhibitors, along with the enhancement of chemotherapeutic cytotoxicity, was shown to be a promising strategy in treating EBV-related lymphomas [173]. Apoptosis induction and the inhibition of EBV-associated gastric carcinoma cell growth were shown by NF-κB inhibitors which downregulate the expression of LMP1 [184]. Furthermore, the contribution of LMP1 to resistance to radiation therapy seems to involve both the activation of the NF-κB and p38-MAPK signalling pathways [184,185]. LMP1-induced immunomodulation of the tumour microenvironment seems to be one of the major components of EBVs’ oncogenic potential, with its various components representing promising therapeutic targets.

## Figures and Tables

**Figure 1 viruses-16-00564-f001:**
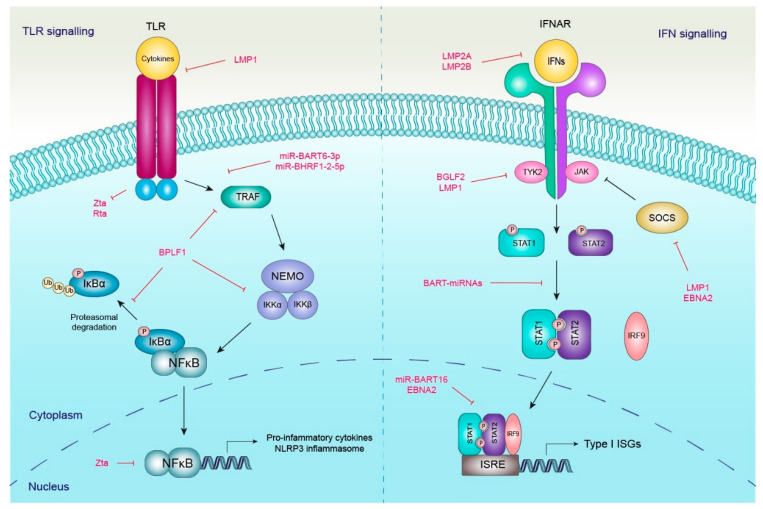
An overview of Epstein–Barr virus (EBV) regulatory strategies for Toll-like receptor (TLR) and interferon (IFN) signalling cascades during the evasion of innate immune responses. Immune response signalling through TLR (left part of the diagram) and IFN (right part) cascades is mainly negatively regulated by both EBV proteins and miRNAs (indicated in red fonts). EBV employs a few tactics for innate immune evasion: (1) downregulation of the expression and activity of TLRs and IFNARs, (2) modulation of type I IFN -STAT pathway and interferon-stimulated gene (ISG) expression, (3) inhibition of NF-κB signal transduction, and (4) repression of the expression of pro-inflammatory cytokines and inflammasomes. See text for further details.

**Figure 2 viruses-16-00564-f002:**
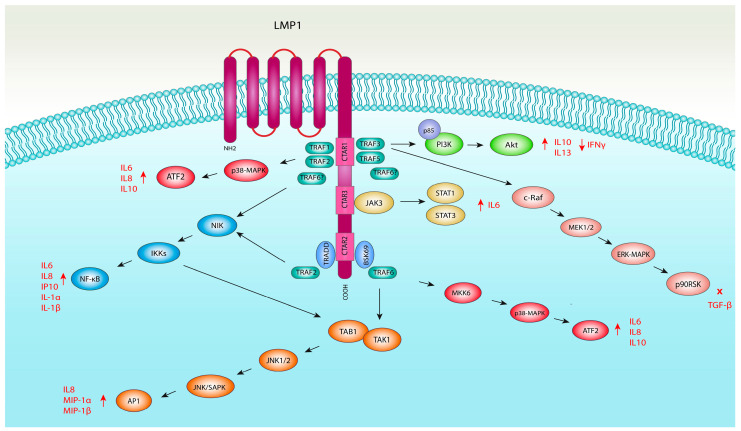
An overview of LMP1-activated signalling pathways included in the induction of cytokines and chemokines. Three C-terminal activating regions (CTARs) were shown to exert the majority of LMP1’s impact on cellular homeostasis by activation of the NF-κB, MAPK (JNK, ERK1/2, p38), JAK/STAT and PI3K/Akt pathways.

## Data Availability

Not applicable.

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
