# Peer review of "Deciphering the Role of Epstein–Barr Virus Latent Membrane Protein 1 in Immune Modulation: A Multifaced Signalling Perspective"

_viruses, 2024, doi:10.3390/v16040564_

Round 1

Reviewer 1 Report (Previous Reviewer 3)

Comments and Suggestions for Authors

The review is out of date, many recent publications are not cited. The information in this review can be found in many published review papers. I don't know that new information this review can provide. The content is also not much related to EBV LMP1. Rather, it mostly focuses on EBV biology

Comments on the Quality of English Language

none

Reviewer 2 Report (Previous Reviewer 2)

Comments and Suggestions for Authors

The authors have well answerd the poits raised by the reviewer

Reviewer 3 Report (New Reviewer)

Comments and Suggestions for Authors

In this review the authors aim to explore the various strategies of evasion of immune defenses of Epstein-Barr virus (EBV)  with a particular focus on the role of the EBV Latent Membrane Protein 1 (LMP1). LMP1 is known to contribute to survival and proliferation of EBV infected B cells and to drive some of the innate anti-viral responses. LMP1 is considered as a constitutively active CD40-ligand receptor that activates mutliple signaling pathways, such as JAK/STAT pathway as well as canonical and non-canonical NF-kappaB signalling pathways. LMP1 has been the focus of a great number of publications. As a consequence the review is very dense. It provides extensive information about what is known of the role and various mechanisms of action of LMP1.

This manuscript is a resubmission of an earlier submission. The following is a list of the peer review reports and author responses from that submission.

Round 1

Reviewer 1 Report

Comments and Suggestions for Authors

The review titled “Deciphering the role of Epstein-Barr virus Latent Membrane Protein 1 in immune modulation: a multifaced signaling perspective” comprehensively summarized the mechanism of LMP1-induced activation of NF-κB, MAPK, JAK-STAT, and PI3K/Akt for immunomodulation through secretion of multiple cytokines. The authors also discussed the significance of immunomodulation for successful viral persistence which would contribute to the understanding of EBV-associated disease pathologies. The present article is well-written and well-structured with proper citations.  I don’t have any major concerns except for a few minor comments to improve the quality of the review article.

Minor Comments:

1-   The resolution of the Figure 2 illustration is very poor and hardly visible/readable. Authors should increase the font size and highlight the secreted cytokines with their role in immunomodulation.

2-   Authors should discuss the role of LMP2A in dampening the reactivity of CD8+ T cells against EBV-infected cells (PMID: 26067064).

3-   LMP1 is also involved in the induction of autophagy (PMID: 18037963). Authors should discuss the divergent role of LMP1-induced autophagy in immunomodulation as autophagy initiates innate immune response by cooperating with pattern recognition receptor signaling for induction of interferon production.

4- The conclusion section is very short. Authors should provide in-depth insight into LMP1-mediated immunomodulation indicating its significance in EBV-associated disease pathologies for developing therapeutic implications.

Comments on the Quality of English Language

 Minor editing of English language required

Reviewer 2 Report

Comments and Suggestions for Authors

In this review “Deciphering the role of Epstein-Barr virus Latent Membrane Protein 1 in  immune modulation: a multifaced signaling perspective” the authors describe the various functions and pathway involved in the expression of the Latent membrane protein 1 (LMP1) expressed during EBV latency and the current knowledge on the functions and interaction of this protein. In the first half of the manuscript the authors introduce a bit of history of the above-mentioned virus and its biology such as its replication, way of infection and pattern of expression during infection and latency. The second half on the other hand is more focused on LMP1 and its interaction with different pathways. The information presented in the manuscript are accurate and well explained.

Comments on the Quality of English Language

Minor editing of English language required

Reviewer 3 Report

Comments and Suggestions for Authors

RE: viruses-2665386

This review aims to summarize the roles of the EBV protein LMP1 in host immune regulation. It is outlined in four sections, introduction, immune response to EBV, specific immune response to EBV, and LMP1 signaling pathway.

The main issue in this manuscript is that most portions are not focused on LMP1. While Section 4 is focused on LMP1, not much of this section is talking about LMP1 immune regulation. It is talking about LMP1 signaling that covers activation of NFκB, p38/AP1, Jak-STAT1, PI3K/Akt pathways. It is not clear how these pathways, which are well documented in many reviews, relate to immune regulation.

·         Section 1. Introduction. EBV structure and replication are introduced. But LMP1 is mainly expressed in latency. Although LMP1 is also expressed in replicative infection, the introduction is not related to LMP1 background.

·         Section 2.  Immune response to EBV infection. There is nothing on LMP1.

·         Section 3. Specific immune response to EBV infection. Little information on how LMP1 is involved in EBV-specific immune regulation.

·         Section 4. This section is focused on LMP1, but it just outlines LMP1 signaling pathway to NFκB, Ap1, Akt activation, which are well known and no new research updates on these pathways. It is not clear how these pathways are involved in immune regulation.

Comments on the Quality of English Language

None